# Vocabulary Fixation Reveals Visual Attention Sink for Hallucination Mitigation in LVLMs

## Abstract

Large Vision-Language Models (LVLMs) show remarkable multimodal progress, but their reliability is undermined by hallucinations, the tendency to generate text that contradicts visual input. Recent work has established a strong link between hallucination and the model's attention to visual tokens. However, the current understanding of the Visual Attention Sink (VAS) phenomenon—where LVLMs persistently assign high attention to uninformative background tokens—remains superficial, leaving both its underlying mechanisms and its connection to the hallucination phenomenon unexplored. In this work, we present the first in-depth analysis of VAS. Using logit lens, we uncover a key property we term **Vocabulary Fixation**: VAS tokens consistently map to a small, fixed set of semantically vacuous words across all layers. Based on this observation, we propose **Vocabulary Fixation-Based Identification (VFI)** to reliably localize visual sink tokens in LVLMs. Furthermore, we establish a strong correlation between VAS and hallucination, and introduce the *Non-Sink Visual Attention Ratio (NVAR)*, a novel metric to precisely identify attention heads critical for mitigating hallucination. Building on this foundation, we propose **Sink-Aware Visual Attention Enhancement (SAVAE)**, a training-free method that adaptively strengthens the attention of these targeted heads to salient visual content during inference. Extensive experiments across multiple LVLMs and benchmarks demonstrate that SAVAE significantly outperforms existing decoding strategies in mitigating hallucination, while introducing no additional computational overhead.

## 1 Introduction

Large Vision-Language Models (LVLMs), which extend large language models (LLMs) with the ability to process visual inputs, represent a major advancement in artificial intelligence (Liu et al., 2023; Chen et al., 2023; Zhu et al., 2023). Nevertheless, LVLMs remain susceptible to the problem of hallucination (Sun et al., 2024b; Zhou et al., 2024; Huang et al., 2024; Bai et al., 2024), where the generated text fails to align with the visual content. Such inconsistencies undermine both the accuracy and reliability of LVLMs in multimodal tasks, thereby limiting their practical deployment.

(a) Average Head Selection  (b) SAVAE Head Selection

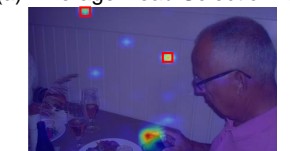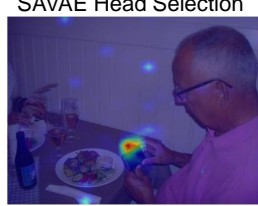

Figure 1: Attention maps for generating "phone". (a) The average of all heads exhibits the attention sink phenomenon, with focus scattered on the background. (b) In contrast, heads selected by our method, SAVAE, concentrate attention precisely on the target object.

Extensive research has focused on mitigating hallucinations in LVLMs, yielding a variety of training-based and training-free interventions (Yu et al., 2024; Leng et al., 2024; Chen et al., 2024; Li et al., 2025). A critical line of inquiry within these efforts has been to analyze the root causes of the phenomenon. **From this research, a key finding has emerged**: insufficient attention to visual tokens during text generation is a primary cause, which has naturally led to a common mitigation

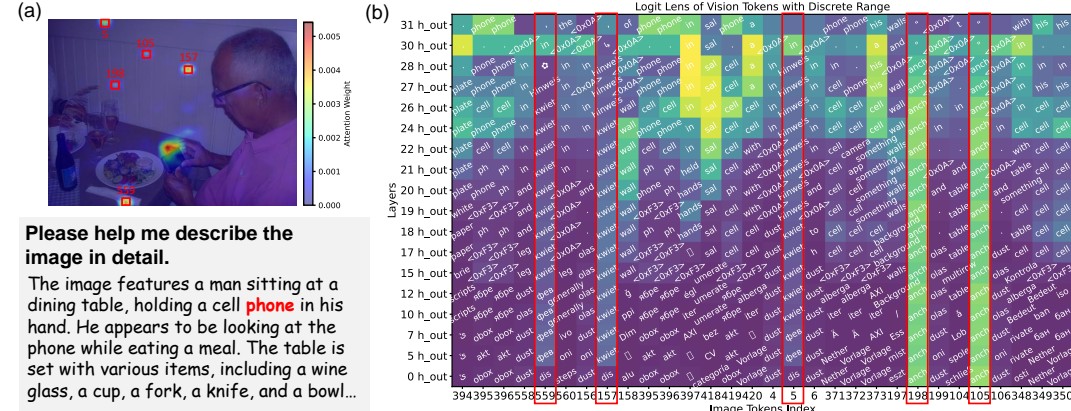

Figure 2: Illustration of the Visual Attention Sink (VAS) phenomenon in LLaVA-1.5 7B. (a) Attention heatmap for the token "phone", where several irrelevant background tokens act as attention sinks (highlighted). (b) Logit lens analysis of the hidden states for the **10 visual tokens to which the current output token assigns its highest attention**, along with their immediate neighbors. This visualization reveals the distinct and non-semantic representations of the identified sink tokens.

strategy of enhancing the visual attention of certain heads during decoding (Jiang et al., 2025; Liu et al., 2024b). However, a significant limitation persists in these approaches: the selection of which heads to modify often relies on heuristics and lacks a principled, quantitative standard for identifying the most critical, hallucination-related heads.

Meanwhile, recent work has revealed the **Visual Attention Sink (VAS)** phenomenon in LVLMs, where the model allocates high attention to specific, often semantically irrelevant, visual tokens. For instance, as shown in Figure 2 (a), when generating the token "phone", the model's second and third highest attention scores are assigned to irrelevant background tokens (559 and 157). Yet, despite such clear empirical evidence, the current understanding of VAS remains superficial. The existing identification method (Kang et al., 2025) is largely a direct adaptation of a text-based approach, leaving the fundamental properties of VAS unexplored. Consequently, the potential role of VAS in critical failure modes like hallucination remains a significant and unaddressed research question.

Motivated by this gap, this paper presents the first in-depth analysis of the VAS mechanism, employing a logit lens approach as exemplified in Figure 2 (b). This analysis uncovers a core property we term **Vocabulary Fixation**: VAS tokens consistently decode to a small set of fixed, meaningless words across all layers. This failure of semantic processing not only explains why these tokens typically manifest as uninformative background patches but also provides the direct foundation for our proposed identification method, **Vocabulary Fixation-Based Identification (VFI)**. Specifically, VFI quantifies this fixation by scoring each visual token based on the frequency with which its decoded representations across all layers fall into this pre-identified set of semantically vacuous words.

Building on this foundation, we investigate the link between VAS and hallucination. Our experiments reveal a strong correlation: greater attention allocated to VAS tokens corresponds to a higher propensity for hallucination. This finding leads us to propose the *Non-Sink Visual Attention Ratio (NVAR)* as a novel criterion for selecting hallucination-related heads, as high-NVAR heads more effectively focus on salient visual information. Based on this, we introduce **Sink-Aware Visual Attention Enhancement (SAVAE)**, a method that mitigates hallucination by selectively strengthening the visual attention of these high-NVAR heads. As illustrated in Figure 1, heads selected by our method exhibit a precise focus on target objects, unlike the scattered attention of average heads.

Extensive experiments confirm SAVAE's superiority: it not only significantly outperforms existing methods in mitigating hallucination across multiple benchmarks, but does so with zero additional computational overhead, demonstrating both state-of-the-art effectiveness and practical efficiency.

Our main contributions are summarized as follows:

- We uncover **Vocabulary Fixation**, a core mechanism of visual attention sinks that explains why they manifest as uninformative background tokens. This key insight directly enables our novel identification method, **VFI**, for reliably localizing them across diverse LVLMs.

- We introduce **SAVAE**, a training-free method that uses a novel metric, **NVAR**, to identify and selectively enhance hallucination-critical attention heads, thereby mitigating hallucinations.
- Through extensive experiments, we demonstrate that SAVAE sets a new state-of-the-art in hallucination mitigation, substantially outperforming prior decoding methods while incurring zero additional computational overhead.

## 2 RELATED WORK

**Reducing Hallucinations in MLLMs.** Mitigating MLLM hallucinations (Li et al., 2023a; Zhou et al., 2023; Liu et al., 2024a) is typically approached via either model fine-tuning (e.g., RLHF-V (Yu et al., 2024)) or training-free interventions at inference time. These interventions include enhancing visual attention (PAI (Liu et al., 2024b), Devils (Jiang et al., 2025)), applying contrastive decoding (VCD (Leng et al., 2024), HALC (Chen et al., 2024)), or steering activations (VISTA (Li et al., 2025)). While PAI and Devils are the most related to our work, their head selection criteria are based on heuristics and lack a principled, quantitative standard—the key gap we address.

**Attention Sink in Language Models.** The attention sink phenomenon, where semantically vacuous tokens attract disproportionate attention due to massive activation patterns, is a well-documented artifact in language models (Xiao et al., 2023; Sun et al., 2024a). While this concept has been extended to the visual domain as the *visual attention sink* (VAS) with initial mitigation efforts (Kang et al., 2025), a deep, mechanistic understanding remains critically lacking. Key questions regarding the origins of VAS, its relationship to text-based sinks, and its direct impact on hallucination are still unexplored. Our work provides the first in-depth investigation aimed at answering these questions.

## 3 FROM VAS TO HALLUCINATION: MECHANISMS AND EVIDENCE

### 3.1 PRELIMINARIES

**Autoregressive Generation in LVLMs.** Large Vision-Language Models (LVLMs) generate responses autoregressively by modeling the conditional probability of the next token. At each timestep $k$, the model predicts the token $y_k$ based on the preceding context, which comprises a sequence of image tokens $\mathcal{I}_v$, a text prompt $\mathcal{I}_t$, and previously generated tokens $\mathcal{I}_o$. These components are concatenated to form a single input sequence $\mathcal{I}$.

**Attention Mechanism in LVLMs.** The core component enabling token interaction is Multi-Head Attention (MHA). Following (Elhage et al., 2021), at layer $\ell$, the representation for a token $x_i^{\ell-1}$ is updated by attending to all previous tokens $X_{\leq i}^{\ell-1} = \{x_0^{\ell-1}, \ldots, x_i^{\ell-1}\}$ as follows:

$$\text{MHA}^{\ell,h}(x_i^{\ell-1}) = \sum_{j \leq i} A_{i,j}^{\ell,h} x_j^{\ell-1} W_{OV}^{\ell,h}, \quad A_i^{\ell,h} = \text{softmax}\left(\frac{(x_i^{\ell-1} W_Q^{\ell,h})(X_{\leq i}^{\ell-1} W_K^{\ell,h})^\top}{\sqrt{D_k}}\right). \quad (1)$$

Here, $W_Q^{\ell,h}, W_K^{\ell,h} \in \mathbb{R}^{D \times D_k}$ are the query and key projection matrices, and $W_{OV}^{\ell,h} \in \mathbb{R}^{D \times D}$ is the output-value projection matrix. The attention weight $A_{i,j}^{\ell,h}$ quantifies the contribution of token $x_j^{\ell-1}$ to the updated representation of token $x_i^{\ell-1}$. Our analysis focuses on the cases where $i \in \mathcal{I}_o$ and $j \in \mathcal{I}_v$, which correspond to the attention paid to visual tokens during the generation process.

**Logit Lens**(nostalgebraist, 2020). To probe the model's internal processing of visual information, we use the Logit Lens method. It maps an intermediate visual hidden state $v_i^\ell$ directly to a distribution over the vocabulary $\mathcal{V}$ by applying the model's final unembedding matrix, $W_{\mathcal{V}} \in \mathbb{R}^{|\mathcal{V}| \times d}$:

$$\mathbf{p}(\mathcal{V}|v_i^\ell) = \text{softmax}(W_{\mathcal{V}} \cdot v_i^\ell) \in \mathbb{R}^{|\mathcal{V}|}, \quad (2)$$

where $p_j(\mathcal{V}|v_i^\ell)$ corresponds to the probability of the $j$-th vocabulary token. We then select the most probable token from this distribution as the textual explanation for the hidden state $v_i^\ell$.

### 3.2 SETUP FOR EXPLORATORY ANALYSIS

Motivated by the methodology of Devils (Jiang et al., 2025), our exploratory analysis is based on a random selection of 500 images from the COCO 2014 validation set (Lin et al., 2014). This dataset

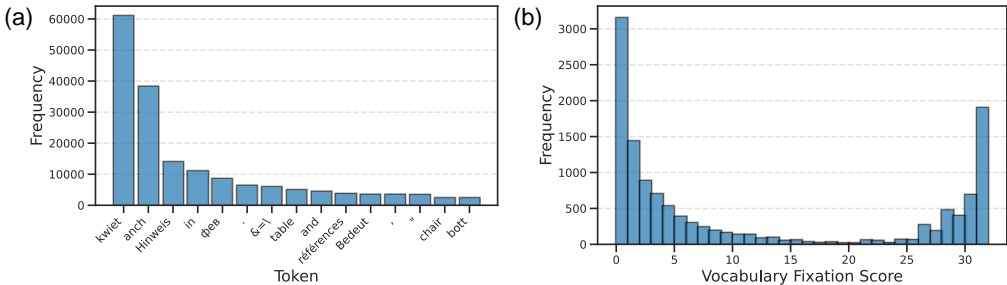

Figure 3: Empirical basis for our VFI method on LLaVA-1.5 7B (500 samples). (a) The distribution of Vocabulary Trajectory Sets for object tokens is shown to be highly concentrated, confirming the Vocabulary Fixation phenomenon. (b) This concentration leads to a distinct U-shaped distribution of Vocabulary Fixation Scores, which provides a clear and principled basis for selecting a threshold $\tau$ to effectively separate normal tokens from VAS tokens.

is chosen for its diversity object categories and providing a rich set of ground-truth annotations for each image. For this analysis, we examine the outputs of LLaVA-1.5 7B and 13B, Shikra-7B, and MiniGPT4-7B. We employ **greedy search** to generate a detailed description for each image, using the prompt: *"Please help me describe the image in detail"*.

By comparing the generated text against the ground-truth annotations, we categorize all mentioned objects into two sets: real object tokens, $\mathcal{O}_{\text{real}}$, and hallucinated object tokens, $\mathcal{O}_{\text{hall}}$. This data provides the empirical foundation for our subsequent investigation into the relationship between VAS and hallucination.

### 3.3 UNDERSTANDING AND DETECTING VAS TOKENS VIA LOGIT LENS

Prior work (Jiang et al., 2025) demonstrates that normal visual tokens follow a structured semantic trajectory through the model's layers, progressing from **Visual Information Enrichment** in the shallow-to-mid layers to **Semantic Refinement** in the mid-to-deep layers, where they acquire clear meaning. Our logit lens analysis in Figure 2 (b), however, reveals that VAS tokens completely defy this productive pattern. Instead of evolving semantically, they consistently map to a small, fixed set of meaningless words across all layers. We define this persistent, non-semantic mapping as the **Vocabulary Fixation of Visual Sink Tokens**.

To examine this phenomenon's generality, we define the **Vocabulary Trajectory Set** for a visual token $v_i$ as the sequence of its most likely decoded word at each layer:

$$\mathcal{V}_T(v_i) = \{\hat{y}_i^1, \hat{y}_i^2, \ldots, \hat{y}_i^L\}, \tag{3}$$

where $\hat{y}_i^\ell$ is the argmax decoding token of the hidden state $\boldsymbol{v}_i^\ell$. We then aggregate these trajectory sets, which are computed for the 10 visual tokens to which each object token (from Section 3.2) assigns its highest attention. The overall distribution on LLaVA-1.5 7B is shown in Figure 3 (a).

This analysis statistically validates the **Vocabulary Fixation** phenomenon. The most frequent tokens decoded from VAS token trajectories are highly concentrated in a small set of recurring, meaningless vocabulary items (e.g., , *kwiet*), corroborating our case study in Figure 2 (b).

This finding suggests that rather than contributing to visual-semantic understanding, **VAS tokens are captured by an internal mechanism that confines them to a semantically inert subspace throughout the LVLM's processing layers**. This mechanism, in turn, provides a compelling explanation for the common observation that VAS tokens are typically located in background regions and are devoid of meaningful semantic content. To demonstrate the generality of this finding, we present the distribution of Vocabulary Trajectory Sets for additional models in Appendix B.1.

### 3.4 VOCABULARY FIXATION-BASED IDENTIFICATION

**Quantifying Vocabulary Fixation.** Based on the Vocabulary Fixation phenomenon observed in Section 3.3, we propose a method to identify VAS tokens. First, we construct a model-specific set,

$\hat{\mathcal{S}}$, containing the most frequent, semantically vacuous tokens from our statistical analysis. For any given visual token $v_i$, we then compute its **Vocabulary Fixation Score**, $f(v_i)$, defined as the number of times its Vocabulary Trajectory Set, $\mathcal{V}_T(v_i)$, contains a token from $\hat{\mathcal{S}}$:

$$f(v_i) = \sum_{\hat{y} \in \mathcal{V}_T(v_i)} \mathbf{1}_{\hat{\mathcal{S}}}(\hat{y}). \tag{4}$$

A visual token $v_i$ is identified as a VAS token if $f(v_i)$ meets or exceeds a predefined threshold $\tau$.

**Parameter Selection.** The parameters $\hat{\mathcal{S}}$ and $\tau$ are determined empirically. The selection of the fixed vocabulary set $\hat{\mathcal{S}}$ is guided by the concentrated nature of the Vocabulary Fixation phenomenon (Figure 3). To ensure high recall, we set its size to 10 for LLaVA-1.5 and MiniGPT-4, manually excluding any tokens with explicit semantic meaning (e.g., "in"). For Shikra, where the fixation is even more pronounced and primarily on the $$ token, we use a more targeted set of size 1.

Once $\hat{\mathcal{S}}$ is defined, we determine the threshold $\tau$. By visualizing the distribution of Vocabulary Fixation Scores, we consistently observe a distinct U-shaped pattern across all models. This holds true for our main example in Figure 3 (b) and is further demonstrated across additional models in Appendix B.1. This pattern creates a natural separation between low-scoring (normal) and high-scoring (sink) tokens, allowing us to intuitively select an effective threshold from the distribution's valley (e.g., $\tau = 23$ for LLaVA-1.5-7B). Crucially, the clarity of this separation ensures that the final identification is not sensitive to minor perturbations of this threshold value.

**Final Identification Rule.** With both the set $\hat{\mathcal{S}}$ and the threshold $\tau$ established, the final identification rule is as follows:

$$\mathcal{I}_{\text{sink}} = \{v_i \in \mathcal{I}_{\text{v}} \mid f(v_i) \geq \tau\}. \tag{5}$$

The complete, step-by-step process for this framework is formalized in Algorithm 1 in Appendix G. To further underscore the superiority of our approach, we also provide a case study in Appendix C that contrasts VFI with the massive activation-based method (Kang et al., 2025).

## 3.5 NVAR: A Metric for Identifying Hallucination-Related Heads

To develop a principled method for selecting hallucination-related heads, we first need a metric that quantifies their focus on meaningful, non-sink visual information. To this end, we introduce the **Non-Sink Visual Attention Ratio** (NVAR), defined as the proportion of a head's total attention budget that is allocated to non-sink visual tokens:

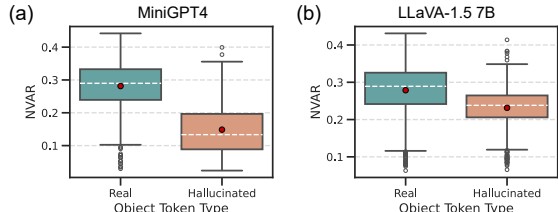

Figure 4: **NVAR distribution of (a) MiniGPT4 and (b) LLaVA-1.5 7B**

$$\text{NVAR}^{(\ell,h)}(y_k) \triangleq \frac{\sum_{v_i \in \mathcal{I}_{\text{v}} \setminus \mathcal{I}_{\text{sink}}} A_{k,i}^{(\ell,h)}}{\sum_{v_i \in \mathcal{I}} A_{k,i}^{(\ell,h)}}. \tag{6}$$

The set of sink tokens, $\mathcal{I}_{\text{sink}}$, is identified using our VFI (Section 3.4). A higher NVAR score thus indicates a head that is more robust against the sink effect and better grounded in visual information.

We validate NVAR by analyzing its statistical relationship with hallucination within a critical subset of heads. Acknowledging the functional specialization of heads (Deiseroth et al., 2023; Zhang et al., 2024; Ge et al., 2024; Zheng et al., 2024), we first select the top 450 heads with the highest mean NVAR scores. Within this pre-selected group, we compare the NVAR distributions for real versus hallucinated object tokens. The results visualized in Figure 4 reveal a stark contrast: **real object tokens are associated with significantly higher NVAR scores, while hallucinated tokens exhibit markedly lower ones**. This provides strong evidence that NVAR is a reliable indicator of factual grounding, making it a principled criterion for identifying the heads most critical to the hallucination phenomenon. This finding holds consistently across additional models, as shown in Appendix B.2.

Furthermore, to provide a deeper characterization of the VAS tokens, and due to space constraints, our analysis of their positional distribution is presented in Appendix B.3.

Table 1: Performance of **SAVAE(Ours)** against baselines. Best results are in **bold**. Pink cells mark potentially unreliable CHAIR scores. Superscripts show the % change vs. the best baseline. †Re-evaluated using the baseline's CHAIR hyperparameters on all benchmarks for consistency.

| Model | Method | CHAIR | | | POPE | | POPE Chat | |
|-------|--------|-------|-------|-----|------|-----|-----------|-----|
| | | CHAIR$_s \downarrow$ | CHAIR$_i \downarrow$ | F1 $\uparrow$ | Acc. $\uparrow$ | F1 $\uparrow$ | Acc. $\uparrow$ | F1 $\uparrow$ |
| LLaVA-1.5-7B | Greedy | 48.2 | 14.2 | 76.4 | 84.8 | 85.5 | 85.5 | 83.4 |
| | PAI | 23.8 | 6.2 | 76.8 | 85.9 | 86.0 | 85.5 | 83.4 |
| | Devils | 27.2 | 7.0 | 76.1 | 85.5 | 85.8 | 87.6 | 86.9 |
| | VISTA† | 15.6 | 5.2 | 67.3 | 56.7 | 63.3 | — | — |
| | SAVAE(Ours) | $18.2^{-23.5\%}$ | $3.8^{-38.7\%}$ | 76.7 | $86.1^{+0.2\%}$ | $86.2^{+0.2\%}$ | $88.0^{+0.5\%}$ | $87.0^{+0.1\%}$ |
| MiniGPT-4-7B | Greedy | 28.2 | 8.8 | 73.7 | **76.8** | 76.6 | 77.7 | 76.9 |
| | PAI | 22.6 | 7.6 | 72.9 | 74.7 | 76.3 | 79.1 | 78.8 |
| | Devils | 21.9 | 7.9 | 71.5 | 72.3 | 75.9 | 79.4 | 78.7 |
| | VISTA† | 18.0 | 4.3 | 68.3 | 66.6 | 74.4 | — | — |
| | SAVAE(Ours) | $21.8^{-0.5\%}$ | $6.9^{-9.2\%}$ | 72.5 | $74.3^{-3.3\%}$ | $77.0^{+0.5\%}$ | $80.2^{+1.0\%}$ | $80.2^{+1.8\%}$ |
| Shikra-7B | Greedy | 56.8 | 14.8 | 75.4 | 80.6 | 81.2 | 76.4 | 78.3 |
| | PAI | 36.1 | 9.8 | 75.4 | **81.3** | 81.1 | **76.5** | 77.5 |
| | Devils | 26.2 | 9.3 | 73.0 | 80.5 | 80.4 | 75.7 | 77.7 |
| | VISTA† | 32.8 | 9.8 | 73.4 | 79.0 | 76.8 | — | — |
| | SAVAE(Ours) | $15.8^{-39.7\%}$ | $5.0^{-46.2\%}$ | 71.8 | $80.2^{-1.4\%}$ | $81.3^{+0.1\%}$ | $76.0^{-0.7\%}$ | $78.5^{+0.3\%}$ |
| LLaVA-1.5-13B | Greedy | 41.6 | 11.1 | 79.3 | **82.6** | 84.5 | 85.4 | 83.2 |
| | Devils | 29.0 | 8.6 | 79.9 | 71.4 | 77.2 | 87.8 | 86.4 |
| | SAVAE(Ours) | $21.8^{-24.8\%}$ | $5.0^{-41.9\%}$ | 79.8 | $82.5^{-0.1\%}$ | $84.7^{+0.2\%}$ | $87.9^{+0.1\%}$ | $86.6^{+0.2\%}$ |

## 4 SINK-AWARE VISUAL ATTENTION ENHANCEMENT

Having established the Non-Sink Visual Attention Ratio (NVAR) as a reliable indicator of factual grounding, we now introduce the **Sink-Aware Visual Attention Enhancement (SAVAE)** method. SAVAE is a training-free, two-stage process that leverages NVAR to identify and strengthen the attention heads most critical for preventing hallucination.

**Stage 1: A Principled Criterion for Head Selection.** Building directly on the findings from Section 3.5, the core of our approach is to select the attention heads that consistently demonstrate a high NVAR. We compute a representative score, $\overline{\text{NVAR}}$, for each head by averaging its NVAR values across the set of real-object tokens ($\mathcal{O}_{\text{real}}$) obtained in Section 3.2:

$$\overline{\text{NVAR}}^{(\ell,h)} = \frac{1}{|\mathcal{O}_{\text{real}}|} \sum_{y_k \in \mathcal{O}_{\text{real}}} \text{NVAR}^{(\ell,h)}(y_k). \tag{7}$$

The $K$ heads with the highest $\overline{\text{NVAR}}$ are then selected to form the target set $T$ for reinforcement.

**Stage 2: Collective Attention Reinforcement.** With the set $T$ of hallucination-critical heads identified, we apply our reinforcement mechanism during inference. The pre-softmax attention scores of each *selected* head $(\ell, h) \in T$ are augmented by a scaled bonus derived from the attention pattern of the entire layer. This allows the most effective heads to be guided by a more holistic signal:

$$\boldsymbol{A}_{k,i}^{(\ell,h)} \leftarrow \boldsymbol{A}_{k,i}^{(\ell,h)} + \alpha \frac{1}{H} \sum_{h'=1}^{H} \left| \boldsymbol{A}_{k,i}^{(\ell,h')} \right|, \quad \forall v_i \in \mathcal{I}_{\text{v}}, \tag{8}$$

where $\alpha$ is the enhancement hyperparameter and $H$ is the total head count per layer. This targeted mechanism selectively boosts the model's focus on effective visual information, thereby directly counteracting the influence of VAS. The complete SAVAE framework is formalized in Algorithm 2.

## 5 EXPERIMENTS

### 5.1 EXPERIMENTAL SETUP

**Models.** We evaluate our method on LLaVA-1.5 (7B, 13B) (Liu et al., 2023), Shikra (7B) (Chen et al., 2023), and MiniGPT-4 (7B) (Zhu et al., 2023) to assess its generalizability and scalability.

**Evaluation Benchmarks.** Following prior work (Liu et al., 2024b; Jiang et al., 2025), we evaluate our method on several hallucination benchmarks. We use **CHAIR** (Rohrbach et al., 2018) for

Table 2: AMBER benchmark results on LLaVA-1.5-7B. Best results are in **bold**. Superscripts show the % change vs. the best baseline.

| Model | Method | Generative | | | | Discriminative | | AMBER |
|---|---|---|---|---|---|---|---|---|
| | | $\text{CHAIR}_i \downarrow$ | Cover $\uparrow$ | Hal $\downarrow$ | Cog* $\downarrow$ | Acc. $\uparrow$ | F1 $\uparrow$ | Score $\uparrow$ |
| LLaVA-1.5-7B | Greedy | 6.0 | 50.6 | 27.4 | 2.8 | 74.8 | 77.6 | 85.8 |
| | PAI | 5.0 | 46.2 | 20.5 | 1.7 | 78.0 | 81.2 | 88.1 |
| | Devils | 3.8 | 46.0 | 20.7 | **1.2** | 77.8 | 81.3 | 88.8 |
| | **SAVAE(Ours)** | $\mathbf{3.6}^{-5.3\%}$ | $\mathbf{51.7}^{+2.2\%}$ | $\mathbf{20.2}^{-1.5\%}$ | $1.3^{+8.3\%}$ | $\mathbf{78.6}^{+0.8\%}$ | $\mathbf{82.7}^{+1.7\%}$ | $\mathbf{89.6}^{+0.9\%}$ |

Table 3: Comparison of head selection strategies on LLaVA-1.5 7B.

| Model | Selecting Strategy | CHAIR | | | POPE | | POPE Chat | |
|---|---|---|---|---|---|---|---|---|
| | | $\text{CHAIR}_s \downarrow$ | $\text{CHAIR}_i \downarrow$ | F1 $\uparrow$ | Acc. $\uparrow$ | F1 $\uparrow$ | Acc. $\uparrow$ | F1 $\uparrow$ |
| LLaVA-1.5-7B | Max Attention | 7.8 | 4.4 | 65.8 | 85.9 | 85.6 | 86.0 | 85.5 |
| | **SAVAE(Ours)** | 18.2 | 3.7 | 76.7 | **86.1** | **86.2** | **88.0** | **87.0** |

captioning evaluation and **POPE** (Li et al., 2023b) for query-based object probing, along with its conversational variant, **POPE-Chat**. To assess out-of-domain performance and fine-grained errors, we also employ **AMBER** (Wang et al., 2023). Further details are provided in Appendix D.1.

**Baselines.** We evaluate SAVAE against several state-of-the-art hallucination mitigation strategies. The most closely related methods are **PAI** (Liu et al., 2024b) and **Devils** (Jiang et al., 2025), which also enhance visual attention but rely on heuristic criteria for head selection. We also compare against an approach with a distinct mechanism: the activation steering method **VISTA** (Li et al., 2025). To ensure a fair comparison, VISTA is evaluated on POPE using its CHAIR hyperparameters, with a detailed justification provided in Appendix D.2.

**Implementation Details.** We set the enhancement hyperparameter $\alpha$ to 0.6 for LLaVA-1.5 7B and MiniGPT-4, and to 0.7 for LLaVA-1.5 13B and Shikra. Across all models, the number of selected attention heads, $K$, is consistently set to 450. Our main experiments employ a greedy decoding strategy, while results using beam search and sampling-based decoding are provided in Appendix E.1. Further details on our hyperparameter settings can be found in Appendix D.3.

## 5.2 MAIN RESULTS

**Superior Performance Across All Benchmarks.** As detailed in Table 1, SAVAE demonstrates a clear superiority over competing methods across all evaluated benchmarks. Our approach excels in both long-form captioning (CHAIR) and short-form query (POPE) settings. For the CHAIR benchmark, SAVAE achieves the best $C_s$ and $C_i$ scores, drastically reducing the $\text{CHAIR}_i$ score by 38.7% on LLaVA-1.5-7B and 46.2% on Shikra-7B relative to the strongest baseline (ignoring two unreliable results from VISTA). Similarly, on both POPE and POPE-Chat, SAVAE consistently yields higher F1 scores. This robust performance underscores the effectiveness and generalizability of our method for hallucination mitigation.

**Scalability to Larger Models.** To assess the scalability of our approach, we applied SAVAE to the LLaVA-1.5 13B model. As shown in Table 1, our method continues to yield significant improvements, confirming its effectiveness is not limited to a specific model size. This result highlights the robust scalability of SAVAE.

**Strong Out-of-Domain Generalization.** To test the generalization of our MSCOCO-derived head selection strategy, we perform an out-of-domain evaluation on the AMBER benchmark using LLaVA-1.5 7B. The results, shown in Table 2, reveal that SAVAE maintains a significant performance gain over the baseline. This robust performance on an unseen data distribution confirms the strong generalization ability of our approach. This finding is further supported by results on additional models, which are detailed in Appendix E.2.

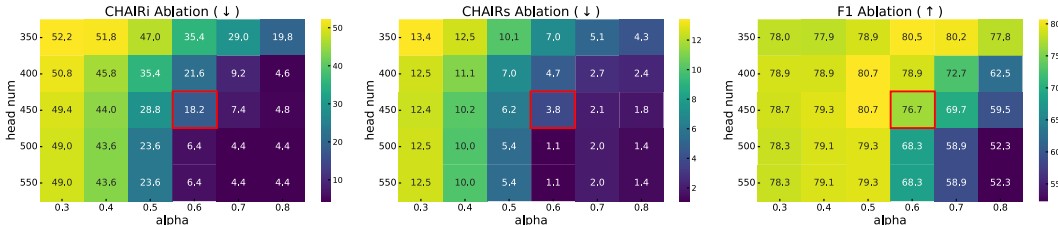

Figure 5: Ablation on hyperparameters $\alpha$ and $K$ for LLaVA-1.5 7B. Red boxes highlight the parameter combinations we used.

Table 4: Ablation study on the sample size for LLaVA-1.5 7B.

| | CHAIR | | | POPE | |
|---|---|---|---|---|---|
| Num | CHAIR$_s$ | CHAIR$_i$ | F1 | Acc | F1 |
| 10 | 18.8 | 3.7 | 76.5 | 86.1 | 86.2 |
| 100 | 18.8 | 3.7 | 76.5 | 85.9 | 86.1 |
| 300 | 18.6 | 4.8 | 76.9 | 85.9 | 86.1 |
| 500 | 18.2 | 3.7 | 76.7 | 86.1 | 86.2 |
| 1000 | 18.2 | 3.7 | 76.7 | 86.1 | 86.2 |

Table 5: Ablation on the sink token penalty factor $\beta$ for LLaVA-1.5 7B.

| | CHAIR | | | POPE | |
|---|---|---|---|---|---|
| $\beta$ | CHAIR$_s$ | CHAIR$_i$ | F1 | Acc | F1 |
| 0.0 | 18.2 | 3.7 | 76.7 | 86.1 | 86.2 |
| 0.1 | 20.2 | 4.5 | 76.8 | 86.1 | 86.2 |
| 0.3 | 19.0 | 4.1 | 77.0 | 86.2 | 86.2 |
| 0.6 | 19.8 | 4.7 | 76.8 | 86.1 | 86.2 |
| 0.9 | 19.2 | 4.7 | 76.9 | 86.2 | 86.3 |

## 5.3 ABLATION STUDIES AND ANALYSIS

**Impact of Hyperparameters $\alpha$ and $K$.** We investigate the impact of hyperparameters $\alpha$ and $K$ by searching over the ranges $[0.3, 0.8]$ and $[350, 550]$, respectively. The results, illustrated for LLaVA-1.5 7B in Figure 5, reveal a clear trade-off. Aggressively increasing $\alpha$ and $K$ effectively suppresses hallucination (lower CHAIR scores) but comes at the cost of reduced generation quality, as indicated by lower F1 scores. Therefore, our final hyperparameters are chosen to strike an optimal balance, maximizing hallucination suppression while preserving the model's fluency and coherence. Ablation studies for our other models are presented in Appendix E.3.

**Superiority of NVAR for Head Selection.** We demonstrate NVAR's superiority by ablating its core component: sink-awareness. As shown in Table 3, selecting heads based only on total visual attention ($\mathcal{I}_{\text{sink}} = \emptyset$) **results in significantly lower F1 scores** across all benchmarks on LLaVA-1.5 7B. This confirms that high raw visual attention is a flawed proxy for effective grounding, as it is easily captured by redundant sink tokens. NVAR's crucial advantage is its ability to filter this noise, isolating heads that truly focus on salient visual content.

**Ablation on Sample Size.** We performed an ablation on the number of samples used for our analysis in Section 3.2. As shown in Table 4, key performance metrics across all benchmarks stabilize once the sample size reaches 500. Therefore, we adopt a sample size of 500 for all experiments, as this provides a robust estimate of performance without incurring unnecessary computational costs.

## 6 FURTHER ANALYSIS

### 6.1 EFFECT OF PENALIZING ATTENTION TO VAS TOKENS

Given the perceived redundancy of VAS tokens, we test the hypothesis that explicitly penalizing their attention could further improve performance. To do this, we introduce a penalty factor $\beta$, where $\beta = 0$ recovers our default SAVAE method. As $\beta$ increases, the enhancement for sink tokens is progressively reduced. The results in Table 5 indicate this strategy offers no significant benefit and can even be detrimental, causing a notable performance drop on CHAIR. This finding, consistent with prior work (Gu et al.), thus justifies our adoption of the simpler, global enhancement strategy.

### 6.2 STATISTICAL CORRELATION BETWEEN VAS AND HALLUCINATION

To provide further statistical evidence linking VAS to hallucination, we analyze the **Visual Attention Sink Ratio (VASR)**, conceptually defined as the proportion of a head's visual attention captured by

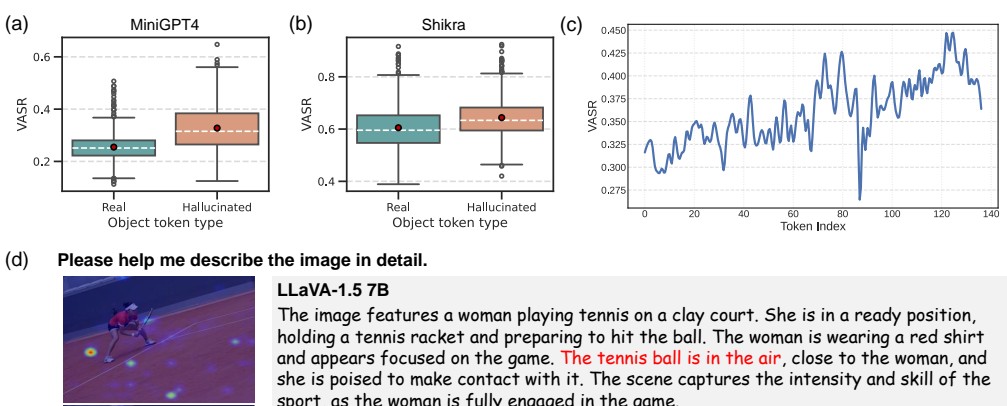

Figure 6: Analysis of the Visual Attention Sink Ratio (VASR) and the effectiveness of SAVAE. (a) and (b) present the statistical distributions of VASR for MiniGPT4 and Shikra. (c) illustrates the evolution of the VASR score during a single generation. (d) A case study demonstrates SAVAE correcting a baseline hallucination by redirecting the model's focus from irrelevant VAS tokens (as shown in the heatmap) back to the salient visual target.

sink tokens ($\mathcal{I}_{\text{sink}}$). Analyzing the top 300 attention heads with the highest total sum of attention to the visual tokens, we find a strong statistical correlation: hallucinated tokens are consistently associated with a significantly higher VASR (Figure 6 (a, b) and Appendix B.2). However, while a strong indicator of hallucination on a token-level, VASR is unsuitable for head selection because its value for any given head is highly context-dependent, fluctuating dramatically based on the token's grounding status, rather than reflecting a stable property of the head itself.

To further illustrate this dynamic, we visualize the evolution of the VASR score over the course of a single generation in a case study shown in Figure 6 (c). As the generation progresses, the VASR steadily increases, indicating that the model's focus gradually shifts towards irrelevant sink tokens, which ultimately precedes the onset of hallucination.

## 6.3 CASE STUDY

A case study in Figure 6 (d) illustrates our method's effectiveness. While the baseline hallucinates that "The tennis ball is in the air", our method provides a factually accurate description. The attention visualizations reveal the mechanism behind this correction: the baseline, when generating **"ball"**, suffers from severe VAS, with attention scattered onto irrelevant background. In contrast, the heads selected by SAVAE demonstrate a precise focus on the target object when generating **"racket"**, thereby preventing the hallucination. Additional case studies are presented in Appendix F.

## 7 CONCLUSION

This paper provides the first mechanistic analysis of visual attention sinks (VAS) in LVLMs, revealing a core property we term **Vocabulary Fixation**, where VAS tokens consistently decode to a fixed set of semantically vacuous words. This discovery provides the foundation for our **Vocabulary Fixation-Based Identification (VFI)** method for reliably localizing these tokens, which in turn enables our primary contribution, **Sink-Aware Visual Attention Enhancement (SAVAE)**. SAVAE leverages our NVAR metric to identify hallucination-critical heads and selectively strengthen their focus on salient visual content. Our experimental results demonstrate that LVLMs can significantly reduce hallucination through principled, sink-aware edits to the attention map at inference time. We believe our work contributes to a deeper, mechanistic understanding of the VAS phenomenon and its link to hallucination, offering a new direction for improving the reliability of LVLMs.

ETHICS STATEMENT

In this paper, we propose a method to analyze and mitigate hallucinations, a key failure mode that undermines the trustworthiness of LVLMs. Our work is intended to be a positive contribution towards developing more reliable and factually grounded AI systems. The core of our analysis, VFI, provides a new tool for auditing and understanding the internal mechanisms of these models, which we believe serves the broader goal of AI safety and transparency. We recognize that any deep analysis of model internals could potentially be used to identify new vulnerabilities; however, our primary contribution, SAVAE, is a defensive mechanism designed to make models more robust. We welcome feedback from the community on further considerations for the responsible development and deployment of this technology.

REPRODUCIBILITY STATEMENT

To ensure full reproducibility, we provide all necessary details in Section 5 and the Appendices. Our experimental setup, including the specific models used and their versions, is detailed in Section 5.1. The datasets, splits, and benchmark configurations for both analysis and evaluation are provided in Appendix D. All hyperparameters for our core contributions, the VFI method ($\hat{S}$ and $\tau$) and the SAVAE method ($\alpha$ and $K$), are specified in Appendix D.3. The complete algorithmic frameworks for VFI and SAVAE are formalized in Appendix G. To facilitate immediate review, we have included the core implementation code in the supplementary materials. Upon acceptance, the full codebase for our analysis and the SAVAE implementation, along with all scripts required to reproduce our results, will be open-sourced under an MIT license.

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

APPENDIX

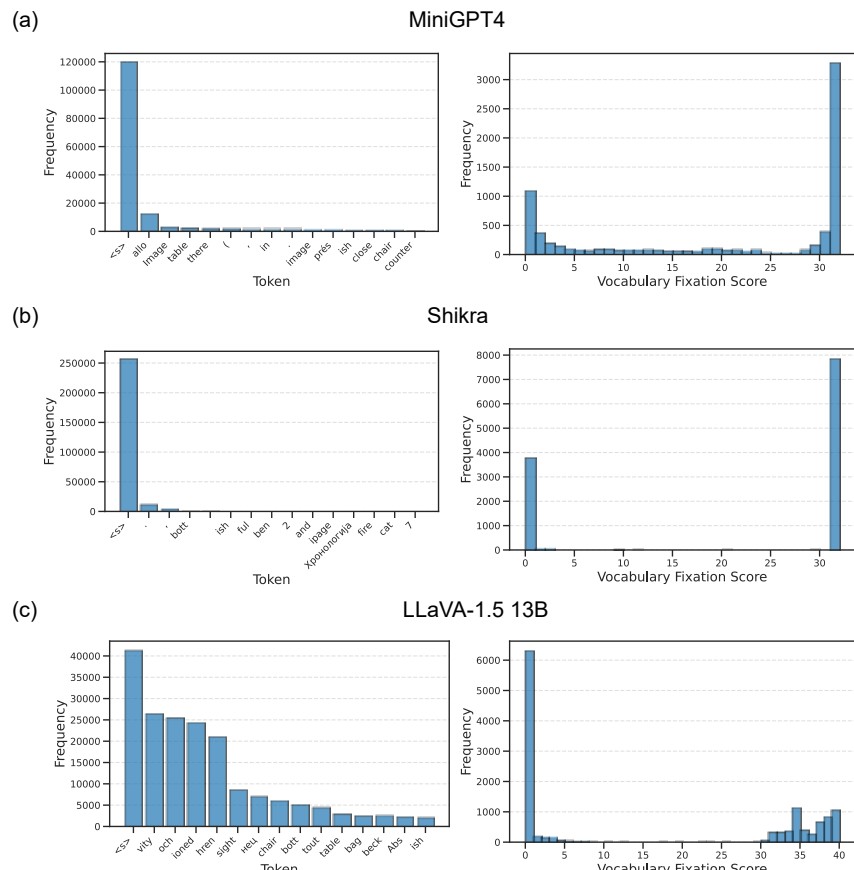

Figure 7: Empirical basis for our VFI method across three different models: (a) MiniGPT4, (b) Shikra, and (c) LLaVA-1.5 13B. For each model, the left plot visualizes the distribution of Vocabulary Trajectory Sets, confirming that the Vocabulary Fixation phenomenon is highly concentrated. The right plot shows the distribution of the resulting Vocabulary Fixation Scores, where the distinct U-shaped pattern provides a clear and principled basis for selecting a threshold $\tau$ to effectively separate normal from VAS tokens.

## A    THE USE OF LARGE LANGUAGE MODELS

Throughout the preparation of this manuscript, large language models were employed exclusively for light stylistic refinement and the occasional grammatical adjustment. Every conceptual insight, analytical thread, and interpretive conclusion emerged from the authors themselves; no algorithmic assistance was solicited for the framing, design, or substance of the work, and full scientific responsibility rests with the human contributors alone.

## B    MORE STATISTICAL RESULTS

### B.1    EMPIRICAL RESULTS FOR VFI ON ADDITIONAL MODELS

Figure 7 presents the empirical results for our VFI method on MiniGPT4, Shikra, and LLaVA-1.5 13B. The results reveal that MiniGPT4 and Shikra exhibit a more pronounced **Vocabulary Fixation** phenomenon compared to the LLaVA-1.5 series, which indicates a stronger underlying VAS effect in these models. This is also reflected in the more heavily populated right tail of their U-shaped Vocabulary Fixation Score distributions.

For all these models, the U-shaped Vocabulary Fixation score distribution provides a clear demarcation between normal and sink tokens. This clear separation allows for a straightforward and robust

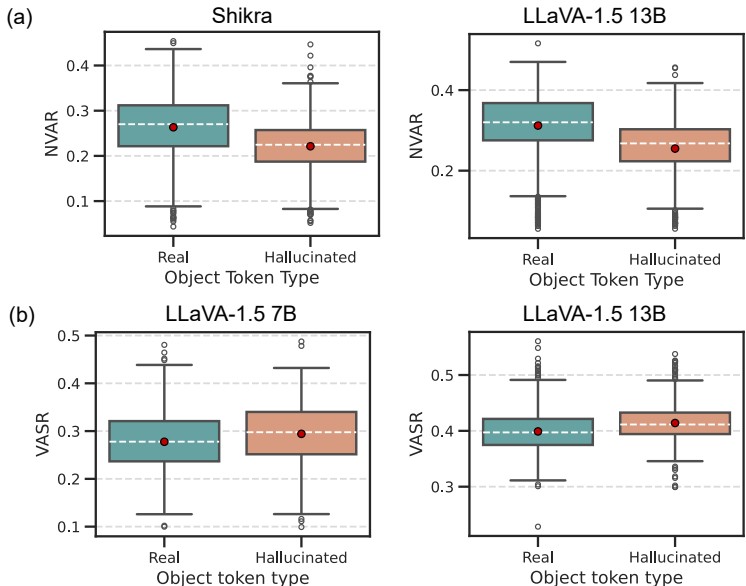

Figure 8: Supporting distributions for NVAR and VASR on additional models. (a) NVAR distributions for Shikra and LLaVA-1.5 13B. (b) VASR distributions for LLaVA-1.5 7B and 13B.

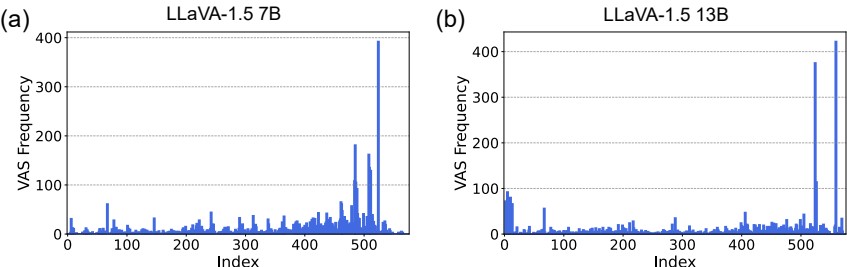

Figure 9: Positional distribution of VAS tokens for LLaVA-1.5 7B and 13B within the 576-token visual sequence. The plots show that VAS tokens are not uniformly distributed, but are instead highly concentrated in specific index ranges. This suggests that the VAS phenomenon may be intrinsically linked to the mechanics of the attention mechanism.

selection of the threshold $\tau$. For instance, we can confidently set $\tau$ to 31, 32, and 31 for MiniGPT4, Shikra, and LLaVA-1.5 13B, respectively, as our analysis shows that the final identification is not sensitive to minor perturbations of these values.

Interestingly, while all these LVLMs are built upon LLaMa-based backbones, their resulting fixation vocabularies are entirely distinct, highlighting the model-specific nature of this phenomenon.

### B.2 NVAR AND VASR DISTRIBUTIONS ON ADDITIONAL MODELS

Figure 8 (a) presents the NVAR distributions for Shikra and LLaVA-1.5 13B. Consistent with our main findings, real object tokens consistently yield higher NVAR scores than hallucinated tokens across these additional models. This result further validates the effectiveness and generalizability of NVAR as a criterion for selecting hallucination-related attention heads.

Part (b) of Figure 8 shows the VASR distributions for LLaVA-1.5 7B and 13B, confirming that hallucinated tokens are associated with higher VASR scores. Furthermore, when considering these results in conjunction with those presented in Figure 6, we can observe the relative severity of the VAS phenomenon across all four tested models. Shikra exhibits the most pronounced effect, while LLaVA-1.5 7B shows the mildest.

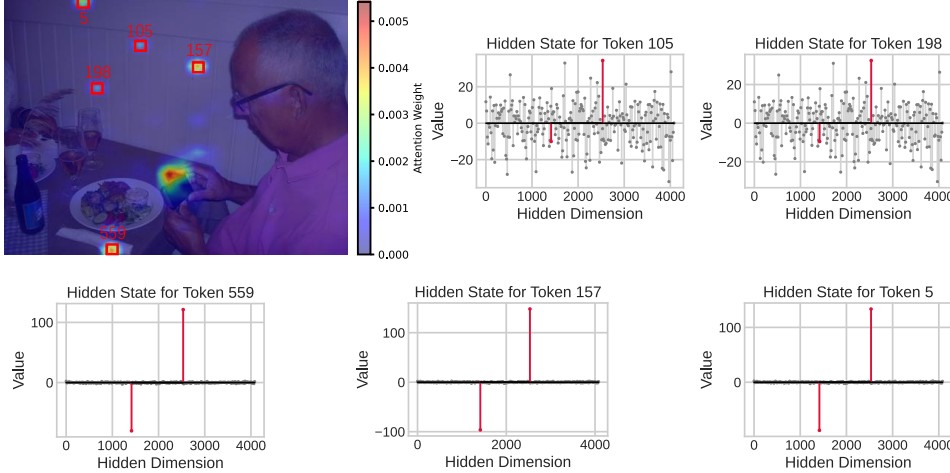

Figure 10: A case study of the hidden state visualization for VAS tokens. The plot shows the hidden state distribution at layer 5 for five prominent VAS tokens identified in an output sequence. The established sink dimensions for LLaVA-1.5 7B, specifically dimensions 1415 and 2533, are highlighted in red for reference.

### B.3 Positional Distribution of VAS Tokens

To further investigate the properties of VAS tokens, we analyzed their positional distribution within the visual sequence. As shown in Figure 9, we present the statistical distribution of the relative indices (from 0 to 575) for VAS tokens in LLaVA-1.5 7B and 13B. The results reveal that VAS tokens tend to be highly concentrated in specific index ranges, rather than being uniformly distributed. This positional bias suggests that the emergence of the VAS phenomenon may be intrinsically linked to the mechanics of the attention mechanism itself.

## C Comparison between VFI and the Massive Activation-Based Method

As there are no direct quantitative metrics to evaluate the performance of VAS identification methods, we provide a qualitative case study in this section to demonstrate the superiority of our VFI method over the massive activation-based approach of (Kang et al., 2025).

We revisit the example from Figure 2, where the model generated the token "phone". Among the top-10 most attended visual tokens, five of them (tokens 5, 105, 157, 198, and 559) were identified as clear VAS tokens. To compare the two identification methods, we visualize the layer-5 hidden state distributions for these five tokens in Figure 10, highlighting the massive activation dimensions $\mathcal{D}_{sink} = \{1415, 2533\}$.

The visualization reveals a critical discrepancy: while several tokens exhibit the phenomenon, **tokens 105 and 198 do not show massive activation in the designated dimensions**. The massive activation method would therefore fail to identify these two tokens. In contrast, our VFI method successfully identifies all five VAS tokens, demonstrating its superior robustness and coverage in detecting the full range of VAS behaviors.

## D Additional Experimental Setups

### D.1 Detailed Benchmark and Evaluation Metrics

**CHAIR** (Rohrbach et al., 2018). The Caption Hallucination Assessment with Image Relevance (CHAIR) metric quantifies hallucination in image captions by comparing generated object mentions against a pre-compiled set of ground-truth objects for each image. An object is considered a hallu-

cination if it is mentioned in the caption but is absent from this ground-truth set. It comprises two scores: instance-level ($\text{CHAIR}_\text{I}$) and sentence-level ($\text{CHAIR}_\text{S}$), calculated as follows:

$$\text{CHAIR}_\text{I} = \frac{|\{\text{hallucinated objects}\}|}{|\{\text{all mentioned objects}\}|}, \tag{9}$$

$$\text{CHAIR}_\text{S} = \frac{|\{\text{captions with hallucinated objects}\}|}{|\{\text{all captions}\}|}. \tag{10}$$

Our evaluation is conducted on 500 randomly sampled instances from the MSCOCO 2014 validation set. To specifically assess long-form generation, we adopt the setup from PAI(Liu et al., 2024b) and Devils(Jiang et al., 2025), generating descriptions with a 'max_new_tokens' of 512 using the prompt: "`Please help me describe the image in detail.`".

**POPE** (Li et al., 2023b). The Polling-based Object Probing Evaluation (POPE) is a benchmark designed within the VQA paradigm to assess object hallucination. It evaluates LVLMs by posing binary questions about object presence, such as "`Is there a <object> in the image?`". The questions are constructed using objects from three distinct sampling strategies to test different aspects of model knowledge: *random* (objects chosen randomly from the dataset), *popular* (frequently occurring objects), and *adversarial* (objects semantically related to those present in the image). *Experimental Setup.* We evaluate on 500 images from the COCO test set, with 6 questions per split for each image, reporting both Accuracy and F1 scores. Furthermore, following (Liu et al., 2024b), to comprehensively examine performance in conversational contexts, we extend the evaluation to include both single-turn and multi-turn dialogues, a setup we term **POPE-Chat**.

**AMBER** (Wang et al., 2023). AMBER is a comprehensive benchmark designed to evaluate multiple facets of hallucination, including object, attribute, and relation errors, across both discriminative and generative tasks. While its discriminative tasks are evaluated using standard metrics (e.g., Accuracy, F1 Score), its generative tasks employ a suite of four specific metrics to assess the quality and faithfulness of model responses. Let $R_\text{obj}$ be the set of objects mentioned in the model's response, $G_\text{obj}$ be the set of ground-truth objects, and $H_\text{obj}$ be a pre-annotated set of common human hallucinations. The generative metrics are defined as follows:

- **CHAIR**: Evaluates the proportion of hallucinated objects among all objects mentioned by the model. Note: This is equivalent to the instance-level CHAIR, and is referred to as $\text{CHAIR}_i$ in our main text.

$$\text{CHAIR} = 1 - \frac{|R_\text{obj} \cap G_\text{obj}|}{|R_\text{obj}|}. \tag{11}$$

- **Cover**: Measures the proportion of ground-truth objects that are correctly mentioned in the model's response (i.e., object recall).

$$\text{Cover} = \frac{|R_\text{obj} \cap G_\text{obj}|}{|G_\text{obj}|}. \tag{12}$$

- **Hal**: A binary metric that indicates whether any hallucination occurred in the response.

$$\text{Hal} = \begin{cases} 1, & \text{if CHAIR} > 0 \\ 0, & \text{otherwise} \end{cases}. \tag{13}$$

- **Cog**: Assesses the similarity between the model's hallucinations and those common to humans.

$$\text{Cog} = \frac{|R_\text{obj} \cap H_\text{obj}|}{|R_\text{obj}|}. \tag{14}$$

Finally, to provide a single, unified measure of performance, AMBER also proposes the **AMBER Score**, which combines the F1 score from discriminative tasks and the CHAIR score from generative tasks:

$$\text{AMBER Score} = \frac{1}{2} \times (1 - \text{CHAIR}_i + \text{F1}). \tag{15}$$

The final reported scores are the average values of these metrics across all queries in the benchmark.

Table 6: Performance of the VISTA method across different models and $\lambda$ values.

| Method | Model | $\lambda$ | CHAIR | | | POPE | |
|---|---|---|---|---|---|---|---|
| | | | CHAIRs | CHAIRi | F1 | Acc | F1 |
| VISTA | LLaVA | 0.01 | 48.4 | 13.2 | 75.9 | 83.1 | 84.6 |
| | | 0.17 | 15.6 | 5.2 | 67.3 | 56.7 | 63.3 |
| | MiniGPT-4 | 0.01 | 35.0 | 9.2 | 69.5 | 76.8 | 77.7 |
| | | 0.10 | 18.0 | 4.3 | 67.3 | 66.6 | 74.4 |
| | Shikra | 0.01 | 55.2 | 15.1 | 74.0 | 82.4 | 82.4 |
| | | 0.12 | 32.8 | 9.8 | 73.4 | 79.0 | 76.8 |

Table 7: Hyperparameter settings.

| Model | $\alpha$ | $K$ | $\tau$ | $|\hat{\mathcal{S}}|$ |
|---|---|---|---|---|
| **LLaVA-1.5-7B** | 0.6 | 450 | 23 | 10 |
| **MiniGPT-4-7B** | 0.4 | 450 | 31 | 10 |
| **Shikra-7B** | 0.7 | 450 | 32 | 1 |
| **LLaVA-1.5-13B** | 0.7 | 450 | 31 | 10 |

## D.2 THE TRADE-OFF BETWEEN THE CHAIR AND POPE BENCHMARKS

In this section, we discuss the importance of maintaining consistent hyperparameters across the POPE and CHAIR benchmarks to provide a comprehensive and fair evaluation of a method's effectiveness. As detailed in Appendix D.1, POPE and CHAIR represent two distinct hallucination scenarios: short-form queries and long-form descriptions, respectively. It is often possible to optimize performance for a single benchmark by aggressively tuning hyperparameters. For example, a higher intervention strength (such as $\alpha$ in our SAVAE method or $\lambda$ in VISTA) can significantly improve CHAIR scores.

However, this often comes at the cost of degraded performance on other benchmarks. Overly aggressive settings can force the model into a conservative generation mode, where it produces only the safest responses, sometimes even leading to repetitive text. This behavior, in turn, typically results in lower scores on benchmarks like POPE. Therefore, we argue that a method's true ability to mitigate hallucination is best demonstrated by robust performance across multiple benchmarks using a single, unified set of parameters.

As an illustration of this trade-off, we present the results for VISTA under different hyperparameter settings in Table 6. The results demonstrate the challenge of achieving a strong balance between the two benchmarks. In our main experiments, all other baseline methods maintain a consistent set of hyperparameters across benchmarks. To ensure a fair and standardized comparison, we adopt this same principle for our evaluation. Consequently, we use the hyperparameter settings proposed for VISTA on the CHAIR benchmark as its single configuration for all evaluations.

## D.3 HYPERPARAMETER SETTINGS

This section provides a cfomprehensive summary of the hyperparameter settings used for each model throughout our experiments. The detailed configurations are presented in Table 7.

# E SUPPLEMENTARY EXPERIMENTS RESULTS

## E.1 PERFORMANCE WITH ALTERNATIVE DECODING STRATEGIES

In this subsection, we present additional experimental results for the beam search and nucleus sampling decoding strategies, which were omitted from the main text due to space constraints. For beam search, we set the beam size to 5, and for nucleus sampling, the temperature is set to 0.5. The de-

Table 8: Performance of **SAVAE** against baselines using beam search decoding. Best results are in **bold**. Pink cells mark potentially unreliable CHAIR scores. Superscripts show the % change vs. the best baseline. †Re-evaluated using the baseline's CHAIR hyperparameters on all benchmarks for consistency.

| Model | Method | CHAIR | | | POPE | | POPE Chat | |
|---|---|---|---|---|---|---|---|---|
| | | CHAIR$_s$ ↓ | CHAIR$_i$ ↓ | F1 ↑ | Acc. ↑ | F1 ↑ | Acc. ↑ | F1 ↑ |
| LLaVA-1.5-7B | Beam Search | 47.6 | 13.0 | 79.0 | 84.7 | 85.4 | 85.3 | 83.2 |
| | PAI | 21.6 | 6.2 | 75.8 | 85.1 | 85.7 | 88.1 | 87.0 |
| | Devils | 29.0 | 6.8 | 80.1 | 84.9 | 85.6 | **88.4** | **87.6** |
| | VISTA† | 9.8 | 5.7 | 54.3 | 56.0 | 63.6 | — | — |
| | **SAVAE(Ours)** | $20.0^{-7.4\%}$ | $5.8^{-6.5\%}$ | 78.7 | $86.0^{+1.1\%}$ | $86.2^{+0.6\%}$ | $88.1^{-0.3\%}$ | $87.1^{-0.6\%}$ |
| MiniGPT-4-7B | Beam Search | 29.0 | 8.9 | 72.9 | **76.9** | 76.9 | 77.9 | 78.0 |
| | PAI | 23.0 | 7.5 | 72.8 | 75.5 | 76.9 | 78.5 | 78.6 |
| | Devils | 20.2 | 6.9 | 71.9 | 67.2 | 74.3 | 79.3 | 79.5 |
| | VISTA† | 15.4 | 4.6 | 67.4 | 64.3 | 73.4 | — | — |
| | **SAVAE(Ours)** | $20.0^{-1.0\%}$ | $6.6^{-4.3\%}$ | 74.3 | $76.6^{-0.4\%}$ | $77.0^{+0.1\%}$ | $79.7^{+0.5\%}$ | $80.4^{+1.1\%}$ |
| Shikra-7B | Beam Search | 56.6 | 14.1 | 77.0 | 81.1 | 81.6 | 77.3 | **78.5** |
| | PAI | 35.4 | 9.2 | 77.1 | **82.0** | 81.3 | **77.4** | 77.3 |
| | Devils | 21.2 | 8.1 | 73.6 | 80.3 | 80.5 | 76.9 | 78.0 |
| | VISTA† | 31.4 | 10.7 | 74.4 | 79.1 | 76.9 | — | — |
| | **SAVAE(Ours)** | $15.0^{-29.2\%}$ | $3.4^{-58.0\%}$ | 72.2 | $81.6^{-0.5\%}$ | $82.0^{+0.5\%}$ | $77.5^{+0.1\%}$ | $78.9^{+0.5\%}$ |

Table 9: Performance of **SAVAE** against baselines using sample as decoding strategy. Best results are in **bold**. Pink cells mark potentially unreliable CHAIR scores. Superscripts show the % change vs. the best baseline. †Re-evaluated using the baseline's CHAIR hyperparameters on all benchmarks for consistency.

| Model | Method | CHAIR | | | POPE | | POPE Chat | |
|---|---|---|---|---|---|---|---|---|
| | | CHAIR$_s$ ↓ | CHAIR$_i$ ↓ | F1 ↑ | Acc. ↑ | F1 ↑ | Acc. ↑ | F1 ↑ |
| LLaVA-1.5-7B | Sample | 48.2 | 15.2 | 73.8 | 83.2 | 84.0 | 85.1 | 83.1 |
| | PAI | 41.6 | 11.3 | 71.7 | 83.5 | 84.2 | 87.0 | 85.9 |
| | Devils | 31.8 | 7.1 | 79.9 | 83.7 | **84.3** | **87.3** | **86.1** |
| | VISTA† | 16.0 | 7.9 | 65.7 | 82.6 | 84.0 | — | — |
| | **SAVAE(Ours)** | $24.8^{-22.0\%}$ | $5.5^{-22.5\%}$ | 77.2 | $84.0^{+0.4\%}$ | $84.3^{\pm0.0\%}$ | $87.0^{-0.3\%}$ | $86.1^{\pm0.0\%}$ |
| MiniGPT-4-7B | Sample | 33.8 | 10.4 | 71.4 | **67.2** | 68.2 | 74.2 | 74.2 |
| | PAI | 28.4 | 12.2 | 69.1 | 65.9 | 68.9 | 75.4 | 75.4 |
| | Devils | 22.2 | **7.7** | 71.9 | 63.0 | 68.0 | 75.0 | 74.5 |
| | VISTA† | 17.4 | 4.8 | 67.7 | 66.5 | 67.3 | — | — |
| | **SAVAE(Ours)** | $22.0^{-0.9\%}$ | $7.8^{+1.3\%}$ | 72.9 | $65.3^{-2.8\%}$ | $69.0^{+0.1\%}$ | $75.7^{+0.4\%}$ | $76.7^{+1.7\%}$ |
| Shikra-7B | Sample | 57.4 | 16.1 | 73.7 | 79.7 | **80.7** | **75.7** | 77.7 |
| | PAI | 41.6 | 11.4 | 72.4 | **80.1** | 80.2 | 75.6 | 76.7 |
| | Devils | 25.0 | 8.8 | 73.3 | 78.6 | 79.4 | 75.2 | 77.3 |
| | VISTA† | 32.6 | 11.0 | 72.5 | 78.7 | 76.4 | — | — |
| | **SAVAE(Ours)** | $18.2^{-27.2\%}$ | $4.6^{-47.7\%}$ | 71.3 | $78.8^{-1.6\%}$ | $80.2^{-0.6\%}$ | $75.5^{-0.3\%}$ | $78.0^{+0.4\%}$ |

tailed results are presented in Table 8 and Table 9, respectively. Overall, the findings are consistent with the conclusions drawn from the greedy decoding experiments in the main text: our method generally outperforms all baseline approaches across these different strategies.

## E.2 AMBER RESULTS ON ADDITIONAL MODELS

In this section, we present the experimental results on the AMBER benchmark for the remaining models, which were omitted from the main text due to space constraints. As shown in Table 10, the overall trend is consistent with the conclusions presented in the main paper: our method SAVAE demonstrates a clear superiority over the baseline approaches.

Table 10: AMBER benchmark results on MiniGPT-4-7B and Shikra-7B. Best results are in **bold**. Superscripts show the % change vs. the best baseline.

| Model | Method | Generative | | | | Discriminative | | AMBER |
|-------|--------|-----------|---------|--------|---------|---------|---------|---------|
| | | CHAIR$_i$ ↓ | Cover ↑ | Hal ↓ | Cog* ↓ | Acc. ↑ | F1 ↑ | Score ↑ |
| MiniGPT-4-7B | Greedy | 15.3 | **63.3** | 65.2 | 11.0 | **64.9** | **65.1** | 74.9 |
| | PAI | 12.3 | 60.8 | 51.3 | 7.2 | 61.4 | 61.3 | 74.5 |
| | Devils | 11.5 | 58.8 | **48.2** | 6.4 | 58.0 | 56.4 | 72.5 |
| | **SAVAE(Ours)** | **11.2**$^{-2.6\%}$ | 61.1$^{-3.5\%}$ | 48.9$^{+1.5\%}$ | **6.1**$^{-4.7\%}$ | 61.4$^{-5.4\%}$ | 61.5$^{-5.5\%}$ | **75.2**$^{+0.4\%}$ |
| Shikra-7B | Greedy | 11.2 | **50.9** | 49.7 | 5.6 | **78.5** | 82.1 | 85.5 |
| | PAI | 7.2 | 49.3 | 34.3 | 3.0 | 78.0 | 82.0 | 87.4 |
| | Devils | 6.7 | 45.3 | 29.5 | **1.6** | 71.1 | 74.1 | 83.7 |
| | **SAVAE(Ours)** | **5.3**$^{-20.9\%}$ | 48.9$^{-3.9\%}$ | **28.4**$^{-3.7\%}$ | 1.7$^{+6.3\%}$ | 78.2$^{-0.4\%}$ | **82.2**$^{+0.1\%}$ | **88.5**$^{+1.3\%}$ |

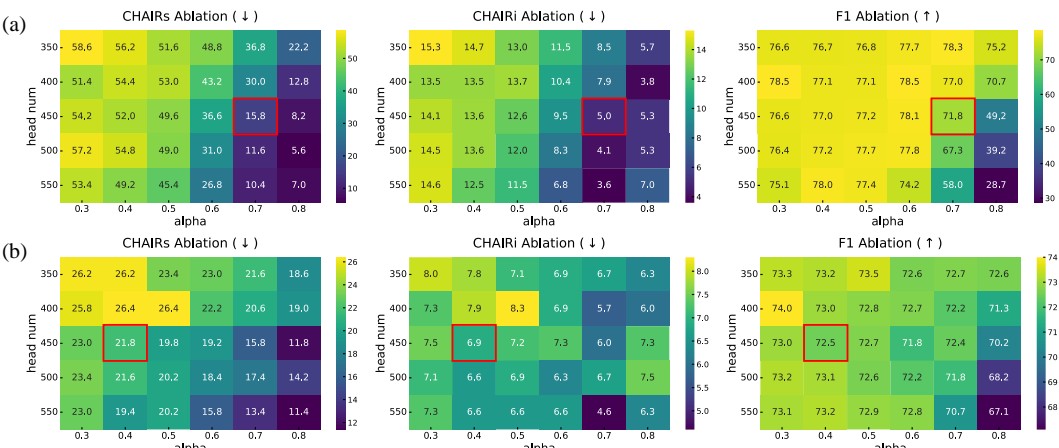

Figure 11: Ablation on hyperparameters $\alpha$ and $K$ for Shikra (a) and MiniGPT4 (b). Red boxes highlight the parameter combinations we used.

### E.3 SUPPLEMENTARY ABLATION RESULTS

In this subsection, we present the supplementary hyperparameter ablation studies for the remaining models, which were omitted from the main text due to space constraints. Specifically, Figure 11 illustrates the results for Shikra and MiniGPT4, which are consistent with the trade-offs discussed in the main paper.

## F ADDITIONAL CASE STUDIES

In this section, we present several additional case studies to qualitatively demonstrate how our SAVAE method concentrates attention on key visual content and effectively mitigates hallucination. Figure 12 visualize four distinct examples. For each example, we contrast the attention map for a hallucinated object token generated by the baseline model with the attention map for a corresponding real object token from our SAVAE-enhanced model. It is important to note a key methodological difference in these visualizations: the baseline maps represent the average attention across **all** heads, whereas the maps for our method show the average attention of the **450 heads selected by SAVAE**.

Additionally, for three of these examples, we provide logit lens visualizations in Figure 13 (following the approach in Figure 2) to further illustrate the behavior of the identified VAS tokens. These visualizations confirm that the model allocates disproportionately high attention to these tokens when a hallucination is produced.

From these examples, we can clearly observe a consistent pattern: when a hallucination occurs, the baseline model's attention is often scattered onto irrelevant VAS tokens. The effect of SAVAE is also made visually evident, as it successfully refocuses the model's attention onto the grounded target object.

**Please help me describe the image in detail.**

**LLaVA-1.5 7B**

The image features a small TV sitting on a wooden desk in a room. The TV is positioned towards the right side of the desk, and it appears to be turned off. The desk is also equipped with a remote control, which is placed near the left side of the TV. In addition to the TV and remote, there are two books on the desk, one located closer to the left side and the other towards the right side. A person can be seen in the background, partially visible on the left side of the image.

**SAVAE (ours)**

The image features a small TV sitting on a wooden desk in a room. The TV is placed on a silver stand, and it is situated next to a window. The television is turned off, and the room appears to be dimly lit. There are several remote controls on the desk. The room has a brown table, and the television is the main focus of the scene.

**Please help me describe the image in detail.**

**LLaVA-1.5 7B**

The image features a woman sitting on a bed in a bedroom. She is wearing a black dress and appears to be looking at something on her laptop, which is placed on her lap. The bed is covered with a white comforter, and there are two pillows on it. In the room, there are several items scattered around. A cat is present, located near the left side of the bed. A cell phone can be seen on the bed, closer to the center. Additionally, there are two bottles, one near the left edge of the bed and the other on the right side. A vase is also visible, placed near the left edge of the bed.

**SAVAE (ours)**

The image shows a woman sitting on a bed with a laptop computer in front of her. She is wearing a black shirt and is looking at the camera. The bed is covered with a white comforter, and the woman is sitting on a white bed. The room is decorated with a red wall and a black and white curtain. The woman is looking at the camera while using her laptop.

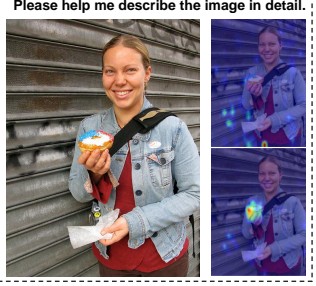

**Please help me describe the image in detail.**

**LLaVA-1.5 7B**

The image features a woman standing outside, holding a donut in her hand. The donut is decorated with red, white, and blue colors, making it visually appealing. The woman is wearing a red shirt and appears to be smiling as she poses with the donut. In the background, there is a backpack placed on the ground, possibly belonging to the woman. The scene seems to be set in an urban environment, with a building visible in the background.

**SAVAE (ours)**

The image features a woman wearing a red shirt and holding a donut with red, white, and blue frosting. She is smiling and holding the donut in front of her face. The woman is wearing a backpack, and she is standing in front of a building. The scene is set in a city, and the woman is holding a donut in her hand.

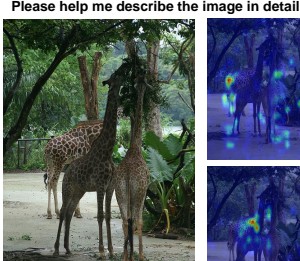

**Please help me describe the image in detail.**

**LLaVA-1.5 7B**

The image features a group of three giraffes standing together in a dirt area. They are positioned close to each other, with one giraffe on the left, another in the middle, and the third on the right. The giraffes are standing under a tree, which provides shade and a natural environment for them. The scene also includes a bench located in the background, possibly for visitors to sit and observe the giraffes. The overall atmosphere of the image is serene and peaceful, showcasing the beauty of these majestic animals in their natural habitat.

**SAVAE (ours)**

The image features a group of three giraffes standing together in a zoo enclosure. They are surrounded by trees and plants, with two of the giraffes eating leaves from a tree. The giraffes are standing in a dirt area, which is their natural habitat. The giraffes are positioned in a way that they are facing each other, creating a sense of unity among them. The scene is set in a forest-like environment, with the giraffes standing near a tree and a fence.

Figure 12: Additional qualitative comparison of attention maps. For each case, we contrast the attention map for a hallucinated token from the baseline model (top row) with a corresponding real object token from our SAVAE-enhanced model (bottom row), demonstrating SAVAE's ability to refocus attention on salient objects.

# G ALGORITHMS

For clarity and to facilitate reproducibility, this section provides the detailed algorithmic procedures for our proposed frameworks. We present the **Vocabulary Fixation-Based Identification (VFI)** framework in Algorithm 1 and the **Sink-Aware Visual Attention Enhancement (SAVAE)** framework in Algorithm 2.

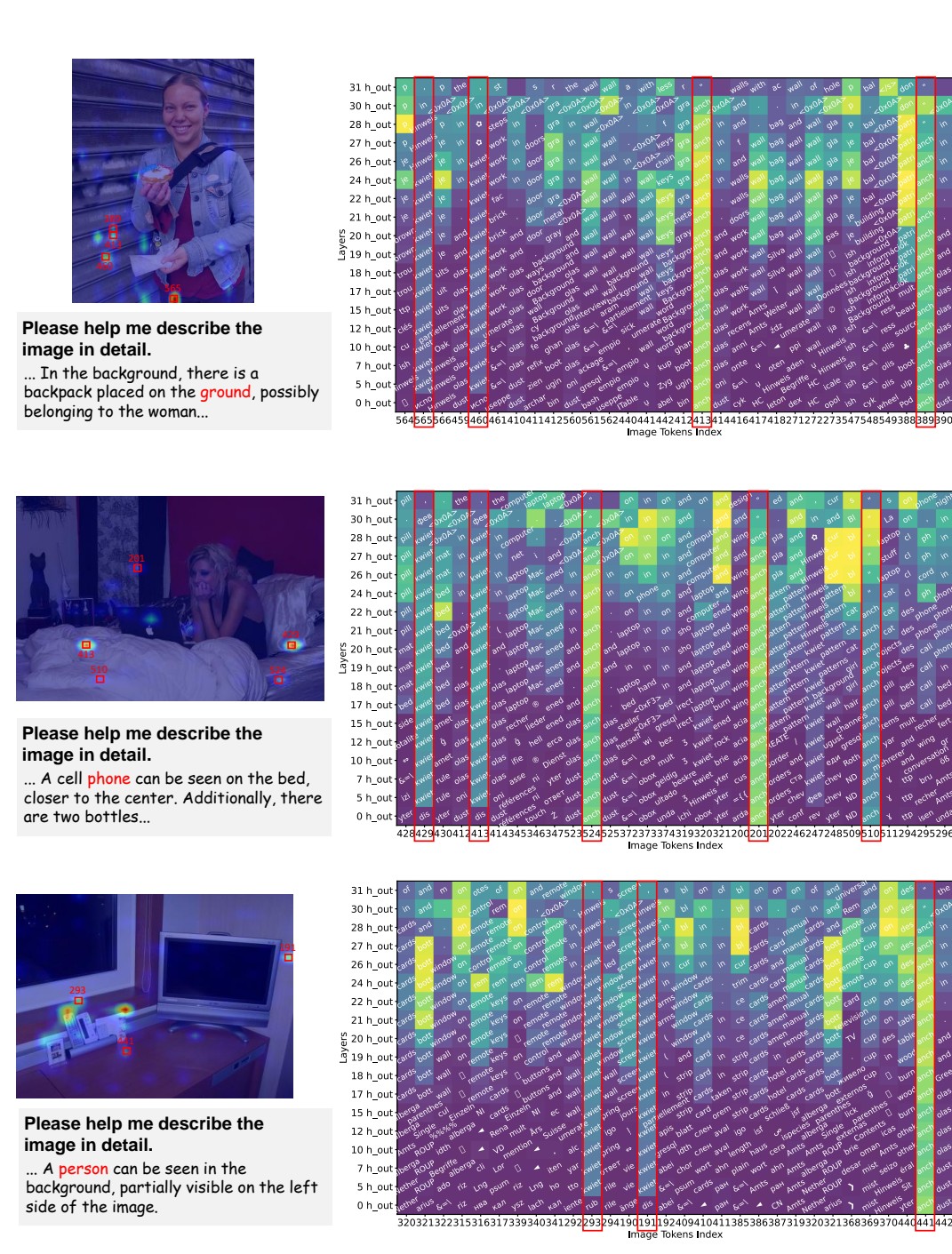

Figure 13: Logit lens visualization for three of the case studies from Appendix F. The plots provide further qualitative evidence that the model allocates disproportionately high attention to the identified VAS tokens during hallucination.

**Algorithm 1** Vocabulary Fixation-Based Identification (VFI) Framework

**Require:** LVLM model $M$; A dataset of images and ground-truth annotations $\mathcal{D}$; TopK_Attended = 10.
**Ensure:** The set of Visual Attention Sink tokens $\mathcal{I}_{\text{sink}}$ for a given image's visual tokens $\mathcal{I}_{\text{v}}$.
1: /* **Phase 1: Data Collection for Analysis** (Corresponds to Sec. 3.2) */
2: $\mathcal{O}_{\text{real}} \leftarrow \emptyset$
3: **for** each image $I$ with annotations $A$ in $\mathcal{D}$ **do**
4:  $D \leftarrow M(I, \text{"Please help me describe the image in detail."})$ {Generate description}
5:  $O_{real\_batch}, O_{hall\_batch} \leftarrow$ CategorizeObjects$(D, A)$
6:  $\mathcal{O}_{\text{real}} \leftarrow \mathcal{O}_{\text{real}} \cup O_{real\_batch}$
7: **end for**
8:
9: /* **Phase 2: Discover Fixed Vocabulary** $\hat{\mathcal{S}}$ **and Threshold** $\tau$ (Corresponds to Sec. 3.3) */
10: $Trajectories \leftarrow []$
11: **for** each real object token $y_k \in \mathcal{O}_{\text{real}}$ **do**
12:  $V_{topk} \leftarrow$ GetTopKAttendedVisualTokens$(M, y_k, \text{TopK\_Attended})$ {Top-10 attended visual tokens}
13:  **for** each visual token $v_i \in V_{topk}$ **do**
14:   $\mathcal{V}_T(v_i) \leftarrow$ GetVocabularyTrajectory$(M, v_i)$ {Decode hidden states}
15:   Append$(Trajectories, \mathcal{V}_T(v_i))$
16:  **end for**
17: **end for**
18:
19: /* Determine $\hat{\mathcal{S}}$ based on concentration */
20: $Freqs \leftarrow$ CalculateTokenFrequencies$(Trajectories)$
21: $\hat{\mathcal{S}} \leftarrow$ SelectTopTokens$(Freqs)$ {Selects top non-semantic tokens after filtering out semantic tokens; Size is adaptive (e.g., 10 for high recall)}
22:
23: /* Compute scores and find threshold $\tau$ from distribution */
24: $Scores \leftarrow []$
25: **for** each trajectory $\mathcal{V}_T$ in $Trajectories$ **do**
26:  $score \leftarrow$ ComputeFixationScore$(\mathcal{V}_T, \hat{\mathcal{S}})$
27:  Append$(Scores, score)$
28: **end for**
29: $\tau \leftarrow$ FindValleyInDistribution$(Scores)$ {Identifies split point in U-shaped distribution}
30:
31: /* **Phase 3: VFI Function for Inference** (Corresponds to Sec. 3.4) */
32: **function** IdentifySinks$(\mathcal{I}_{\text{v}}, M, \hat{\mathcal{S}}, \tau)$ {$\hat{\mathcal{S}}, \tau$ are pre-computed in Phase 2}
33: $\mathcal{I}_{\text{sink}} \leftarrow \emptyset$
34: **for** each visual token $v_i \in \mathcal{I}_{\text{v}}$ **do**
35:  $\mathcal{V}_T(v_i) \leftarrow$ GetVocabularyTrajectory$(M, v_i)$
36:  $f(v_i) \leftarrow$ ComputeFixationScore$(\mathcal{V}_T(v_i), \hat{\mathcal{S}})$
37:  **if** $f(v_i) \geq \tau$ **then**
38:   $\mathcal{I}_{\text{sink}} \leftarrow \mathcal{I}_{\text{sink}} \cup \{v_i\}$
39:  **end if**
40: **end for**
41: **return** $\mathcal{I}_{\text{sink}}$
42: **end function**

---

**Algorithm 2** Sink-Aware Visual Attention Enhancement (SAVAE) Framework

---

**Require:** LVLM model $M$; A dataset of images and ground-truth annotations $\mathcal{D}$; VFI function 'IdentifySinks'; Hyperparameters $K, \alpha$.
**Ensure:** A modified text generation process with reduced hallucination.
  1: /* **Phase 1: Offline Identification of Hallucination-Related Heads** */
  2: **function** SelectHallucinationHeads($M, \mathcal{D}, K$)
  3:   $\mathcal{O}_{\text{real}} \leftarrow$ CollectRealObjectTokens($M, \mathcal{D}$) {As per Sec. 3.2}
  4:   $\overline{\text{NVAR}} \leftarrow$ InitializeMatrix($L, H, \text{zeros}$) {$L, H$: num layers, heads}
  5:   /* Iterate through each token first for efficiency */
  6:   **for** each real object token $y_k \in \mathcal{O}_{\text{real}}$ **do**
  7:     $I_{\text{v}} \leftarrow$ GetCorrespondingVisualTokens($y_k$)
  8:     $\mathcal{I} \leftarrow$ GetFullContext($y_k$) {**Get all context tokens (text+vision)**}
  9:     $\mathcal{I}_{\text{sink}} \leftarrow$ IdentifySinks($I_{\text{v}}, M$) {**From VFI Framework, done once per token**}
10:     $A_{\text{all\_heads}} \leftarrow$ GetAttentionForAllHeads($M, y_k$) {Get attention tensor for step k}
11:     **for** each head $(\ell, h)$ from $(1, 1)$ to $(L, H)$ **do**
12:       $A_{\ell,h} \leftarrow A_{\text{all\_heads}}[\ell, h]$
13:       $N_{numerator} \leftarrow \sum_{v_i \in I_{\text{v}} \setminus \mathcal{I}_{\text{sink}}} A_{\ell,h,i}$
14:       $N_{denominator} \leftarrow \sum_{v_i \in \mathcal{I}} A_{\ell,h,i}$
15:       NVAR\_score $\leftarrow N_{numerator}/N_{denominator}$ {Eq. 6}
16:       $\overline{\text{NVAR}}_{\ell,h} \leftarrow \overline{\text{NVAR}}_{\ell,h} +$ NVAR\_score
17:     **end for**
18:   **end for**
19:   /* Finalize the average scores */
20:   $\overline{\text{NVAR}} \leftarrow \overline{\text{NVAR}}/|\mathcal{O}_{\text{real}}|$ {Eq. 7}
21:   $T \leftarrow$ TopK($\overline{\text{NVAR}}, K$) {Select top-K heads}
22:   **return** $T$
23: **end function**
24: /* **Phase 2: Online Attention Enhancement during Inference** */
25: **function** SAVAE\_Attention\_Forward($A_{original}, T, \alpha$)
26: {This function modifies the pre-softmax attention tensor $A_{original}$ at each generation step.}
27: $L, H \leftarrow$ GetModelDimensions()
28: $A_{enhanced} \leftarrow A_{original}$ {Initialize with original attention}
29: /* Iterate through all layers to apply the enhancement */
30: **for** $\ell = 1$ to $L$ **do**
31:   /* Calculate the collective reinforcement bonus for the current layer $\ell$ */
32:   $Bonus_\ell \leftarrow$ InitializeVector($size = \text{num\_visual\_tokens}$)
33:   **for** $h' = 1$ to $H$ **do**
34:     $Bonus_\ell \leftarrow Bonus_\ell + |\text{AttentionScores}(A_{original}, \ell, h')|$
35:   **end for**
36:   $Bonus_\ell \leftarrow (\alpha/H) \cdot Bonus_\ell$
37:   /* Apply bonus ONLY to selected heads in this layer for their visual attention */
38:   **for** $h = 1$ to $H$ **do**
39:     **if** $(\ell, h) \in T$ **then**
40:       **for** each visual token $v_i$ **do**
41:         $A_{enhanced}[\ell, h, i] \leftarrow A_{enhanced}[\ell, h, i] + Bonus_\ell[i]$ {Eq. 8}
42:       **end for**
43:     **end if**
44:   **end for**
45: **end for**
46: **return** $A_{enhanced}$
47: **end function**

---

