# OpenReview forum: "Vocabulary Fixation Reveals Visual Attention Sink for Hallucination Mitigation in LVLMs"
_ICLR.cc/2026/Conference — ICLR 2026 Conference Withdrawn Submission_

### Official Review · Reviewer_PYSW · 2025-10-19

**Soundness:** 3
**Presentation:** 2
**Contribution:** 2
**Rating:** 2
**Confidence:** 5

**Summary:**

This paper focuses on the phenomenon of visual attention sinks (VAS) in large multimodal language models (LVLMs)—the phenomenon in which the model overattentions certain semantically nonsensical background visual tokens during generation, leading to hallucinations. The authors systematically reveal the underlying mechanism of VAS: vocabulary fixation, where these sink tokens are stably mapped to a small set of meaningless fixed words (e.g., <s>, kwiet) across all network layers. Based on this, the authors propose:

**Strengths:**

The core insight of this paper is the concept of "lexical solidification." Through logit lens analysis, the authors for the first time link the semantic inertness of VAS tokens to the dynamics of the model's internal representations, explaining why these tokens often appear in background regions and providing a theoretical basis for recognition.

**Weaknesses:**

1.The baseline LVLMs used in this paper (LLaVA-1.5, MiniGPT-4, Shikra) are relatively outdated. It is recommended that the authors evaluate the generalization and effectiveness of AAI on more recent baseline LVLMs, such as QwenVL2.5/3, InternVL, more dataset:MME,ScienceQA,GQA....

2.The authors claim that the fundamental properties of VAS remain unexplored. However, to the best of my knowledge, numerous studies have already investigated VAS and its relationship with hallucination, such as [Devils, CVPR 2025], [EAH, EMNLP 2025], [Farsight, CVPR 2025], [TAME, ICLR 2025], [FastV, ECCV 2024], [Clearsight, CVPR 2025], and [SEE WHAT YOU ARE TOLD, ICLR 2025]. It is necessary for the authors to clarify the limitations of these prior works and explicitly distinguish their proposed method from them.

3.The logit lens technique was applied in [Devils, CVPR] to analyze hallucination-related tokens, and it is evident that this paper follows the same approach (as shown in Figure 1(b)). Why is this not acknowledged or cited in Section 3.1?

4.The methodology section requires improved clarity. For example, the variable l in Equation (3) is not defined. The statement “VAS tokens are captured by an internal mechanism that confines them to a semantically inert subspace throughout the LVLM's processing layers” is also confusing and should be clarified.

5.VAS tokens also appear in correct regions in Figure 2—what is the Vocabulary Fixation behavior of these correct VAS tokens? In addition, a comparison of Figure 2(b) before and after AAI intervention should be provided.

6.The paper identifies the Vocabulary Fixation phenomenon by analyzing the decoding trajectories of selected VAS tokens; however, several issues are overlooked. First, since VAS tokens are defined based on attention scores, the correlation between Vocabulary Fixation scores and attention scores remains unclear. Second, the Vocabulary Fixation behavior of non-VAS tokens is not analyzed. Third, existing VAS token identification methods such as TAME, EAH, and SEE WHAT YOU ARE TOLD appear more intuitive and reliable. Finally, the authors determine the threshold hyperparameter for VAS token selection based on the U-shaped distribution of Vocabulary Fixation scores, but this approach lacks rigorous experimental validation and ablation analysis.

7.The design of SAVAE lacks consideration of different attention layers, while prior works such as FastV and EAH have demonstrated that the attention distribution over image tokens varies significantly across layers.

8.The proposed method shows a significant reduction in the CHAIRs score in Table 3, which warrants further attention and discussion. In addition, the experimental results on CHAIR and POPE do not report the recall scores.

**Questions:**

see weakness

---

### Official Review · Reviewer_mRf6 · 2025-10-27

**Soundness:** 2
**Presentation:** 3
**Contribution:** 2
**Rating:** 4
**Confidence:** 3

**Summary:**

This paper analyzes the LVLM hallucination problem and proposes a new perspective to mitigate it. The authors deeply investigate the Visual Attention Sink (VAS) phenomenon, where the model places high attention on uninformative background tokens. In this process, they discover a core phenomenon called Vocabulary Fixation (VF). This is a mechanism where specific visual tokens are repeatedly mapped to the same non-semantic words across all layers, acting as one of the root causes of hallucination.
Based on this insight, the authors propose three components:
1. VFI (Vocabulary Fixation-Based Identification): Utilizes the Vocabulary Fixation phenomenon to quantitatively detect visual sink tokens.
2. NVAR (Non-Sink Visual Attention Ratio): Quantitatively identifies attention heads that focus on meaningful visual information.
3. SAVAE (Sink-Aware Visual Attention Enhancement): A training-free method that mitigates hallucination by enhancing the attention of key heads (selected by NVAR) during inference.
The main contributions of the paper are as follows: First, the discovery of a novel mechanism, Vocabulary Fixation, that explains LVLM hallucination. Second, the establishment of quantitative criteria (VFI and NVAR) to identify hallucination-related tokens and heads. Third, the improvement of the model's visual coherence and reliability without additional training or computational overhead through SAVAE.

**Strengths:**

1. Strong Analysis and Motivation: The analysis related to identifying the cause of hallucination, such as the Vocabulary Fixation phenomenon, is diverse and sound.
2. Clear Narrative Flow: The storyline from motivation to methodology to experiments is natural and well-structured.
3. Effective Visualization: The use of appropriate and varied visualization methods makes it easy to understand the motivation and the effect of the proposed method.

**Weaknesses:**

1. Experimental Aspect: The experiments demonstrating hallucination mitigation are insufficient.
- The motivation experiment (the VAS phenomenon itself) is only shown for the LLaVA model. (Figure 6 appears to show the effect of SAVAE and the validity of the VASR metric rather than the VAS phenomenon itself).
- Although the method was applied to various models/benchmarks to demonstrate hallucination mitigation, the substantial improvements are prominent mainly in the LLaVA model family and the CHAIR benchmark.
- There is no analysis of the impact on general performance, such as performance on other downstream tasks (e.g., VQA, Retrieval).
2. Theoretical Aspect: The discovery of the Visual Attention Sink itself is not entirely novel. While calculating it as an evaluation metric is new, the phenomenon itself has been observed in prior work.

**Questions:**

1. Could the authors provide evidence (e.g., a result table or graph) showing the VAS phenomenon (Figures 2, 3) is confirmed in models other than LLaVA?
2. Have the authors analyzed the impact of their method on the downstream performance of general vision-language tasks (e.g., VQA, Retrieval, etc.)?
3. If the VAS phenomenon (i.e., "sink visual tokens" always being decoded to the same vocabulary at the same spatial location/background patch across different images or prompts) is due to a structural bias within the model itself, is there evidence to show that correcting this bias directly reduces the VAS phenomenon, not just the hallucination mitigation performance? Specifically, are there experiments that confirm a direct correlation with the reduction of the VAS phenomenon?

---

### Official Review · Reviewer_UK6u · 2025-10-29

**Soundness:** 3
**Presentation:** 3
**Contribution:** 2
**Rating:** 4
**Confidence:** 4

**Summary:**

The paper investigates hallucination in large vision–language models (LVLMs) by analysing how attention is distributed over visual tokens. Through a logit‑lens analysis of LLaVA‑1.5, MiniGPT‑4 and Shikra, the authors identify a visual attention sink (VAS) phenomenon: certain image tokens draw disproportionately high attention yet consistently decode to a small, semantically meaningless vocabulary across all layers. This “vocabulary fixation” insight leads to a Vocabulary Fixation‑Based Identification (VFI) method that detects sink tokens by counting how often a token’s decoded trajectory hits a set of frequent, vacuous words. Building on this, the authors propose the Non‑Sink Visual Attention Ratio (NVAR) to quantify how much an attention head focuses on non‑sink tokens. Heads with high NVAR are presumed more grounded and are selected for reinforcement. The final contribution is Sink‑Aware Visual Attention Enhancement (SAVAE), a training‑free inference‑time method that strengthens the pre‑softmax attention of the top‑NVAR heads by adding a scaled bonus from the mean head attention. Experiments on CHAIR, POPE/POPE‑Chat and AMBER show that SAVAE significantly reduces hallucination metrics and often improves F1/accuracy compared with greedy decoding and prior mitigation methods like PAI, Devils and VISTA. SAVAE also generalizes across models (7B and 13B) and out‑of‑domain benchmarks.

**Strengths:**

S1. Identifies Vocabulary Fixation as the underlying cause of visual attention sinks

S2. SAVAE is training-free, incurs zero additional computational overhead, and is straightforward to deploy

S3. The paper is generally well‑structured.

**Weaknesses:**

W1. VFI requires constructing a model‑specific set of “vacuous” tokens ˆS and selecting a threshold tau based on a U‑shaped distribution. Although the authors argue that the separation is clear, the method still involves manual filtering of semantically meaningful tokens and tuning tau on validation data, which may not generalize across languages or domains.

W2. The analysis focuses on a few LVLMs, all built on similar base architectures (LLaVA, MiniGPT‑4, Shikra). It is unclear how vocabulary fixation and NVAR behave on recent models such as GPT‑4o or Gemini, or on language‑only models. The tasks are primarily captioning and object‑query benchmarks; the effect on dialogue, visual reasoning or instruction‑following tasks is not studied.

W3. Computing NVAR involves distinguishing real vs. hallucinated object tokens using MS‑COCO annotations. This may not be feasible in practice or for arbitrary inputs, and the paper does not discuss how SAVAE would be applied without labelled data.

W4. While SAVAE often improves F1, there are small decreases in accuracy in some settings (e.g., MiniGPT‑4 results show a slight drop in POPE‑Acc). It would be useful to analyze qualitative outputs to see whether the method biases captions toward shorter or more generic descriptions.

W5. The reinforcement mechanism boosts attention on selected heads by adding an averaged layer signal. It remains unclear whether this negatively affects capabilities unrelated to hallucination, such as reasoning, multi‑step dialogue or safety.

W6. Missing important prior work [1]. The observation that VAS tokens fixate on a small set of meaningless words across layers is similar to the observation in [1].

[1] Don't Miss the Forest for the Trees: Attentional Vision Calibration for Large Vision Language Models, ACL 2025

**Questions:**

Q1. How robust is the VFI method if the set of vacuous tokens ˆS is defined automatically (e.g., via a clustering criterion) rather than manual selection? Would a mis‑specified ˆS degrade NVAR estimation or head selection?

Q2. Have you tested SAVAE on visual question answering or multimodal dialogue datasets beyond CHAIR/POPE? Do the head selections generalize when the task prompt differs significantly from a captioning prompt?

Q3. SAVAE’s head ranking uses NVAR computed over real‑object tokens identified via ground‑truth labels. How would you apply your method when such labels are unavailable? Could you approximate NVAR using unsupervised cues?

---

### Official Review · Reviewer_LG9y · 2025-10-30

**Soundness:** 3
**Presentation:** 2
**Contribution:** 2
**Rating:** 4
**Confidence:** 3

**Summary:**

This paper addresses the critical hallucination problem in Large Vision-Language Models (LVLMs) by focusing on the Visual Attention Sink (VAS) phenomenon and bringing up training-free solution. The authors’ core contributions include: (1) Uncovering Vocabulary Fixation, showing VAS tokens consistently maps to a small set of semantically vacuous words across all layers; (2) Proposing Vocabulary Fixation-Based Identification to localize VAS tokens; (3) Introducing standard metric (the Non-Sink Visual Attention Ratio, NVAR) to select hallucination-critical attention heads; (4) Developing SAVAE, a training-free method that enhances these heads’ attention to salient visual content during inference. The paper proves the vocabulary fixation phenomenon by designed experiments, and validates SAVAE on multiple models and benchmarks.

**Strengths:**

1. The discovery of the Vocabulary Fixation phenomenon is insightful and sheds light on possible mechanisms underlying hallucinations in LVLMs.

2. The proposed SAVAE method is a effective, training-free approach that helps mitigate hallucinations in LVLMs.

3. The paper presents a reasonably broad empirical evaluation across several LVLMs and benchmarks, which supports the claimed generality of SAVAE.

4. The paper is well written and easy to follow.

**Weaknesses:**

1. The definition of *Vocabulary Fixation* and its detection via VFI are only supported by descriptive statistics and qualitative visualizations. There is no quantitative evaluation (e.g., precision/recall vs. ground-truth or baseline) proving that VFI correctly identifies sink tokens. This is the base for the entire paper so it's necessary to provide more quantitative evidence.

2. The fixed vocabulary ( \hat{S} ) and threshold τ are manually chosen and filtered, yet robustness to these design choices is not analyzed. Reproducibility and generality are unclear.

3. The paper does not test whether SAVAE’s improvements actually stem from targeting Vocabulary Fixation. Ablation experiments (e.g., applying SAVAE to non-sink tokens vs sink tokens) are missing.

**Questions:**

1. Can you provide quantitative evaluation of VFI (precision/recall/F1) against manually annotated or proxy ground-truth sink tokens?

2. How sensitive are your results to the choice and size of the fixed vocabulary ( \hat{S} ) and the threshold τ?

3. Have you tested SAVAE with random or non-sink vocabularies to confirm that its gains specifically result from mitigating fixation?

---

### Note · Authors · 2025-11-19

I have read and agree with the venue's withdrawal policy on behalf of myself and my co-authors.